# VIDEOLIGHTS: A CROSS-MODAL CROSS-TASK TRANSFORMER MODEL FOR JOINT VIDEO HIGHLIGHT DETECTION AND MOMENT RETRIEVAL

## ABSTRACT

Video Highlight Detection and Moment Retrieval (HD/MR) are essential in video analysis. Recent joint prediction transformer models often overlook cross-task dynamics and video-text alignment. We propose **VideoLights**, a novel HD/MR framework addressing these limitations through: **(i)** Convolutional Projection and Feature Refinement modules with an intermodal alignment loss for better video-text feature alignment. **(ii)** Bi-Directional Cross-Modal Fusion network for strongly coupled query-aware clip representations. **(iii)** Uni-Directional joint-task feedback mechanism enhancing both tasks through correlation. In addition, we introduce hard positive/negative losses for adaptive error penalization and improved learning. Our approach includes intelligent pretraining and finetuning using synthetic data and features from various encoders. Comprehensive experiments on QVHighlights, TVSum, and Charades-STA benchmarks demonstrate state-of-the-art performance.

## 1 INTRODUCTION

The surge in digital devices, platforms, and internet usage has led to abundant online video content (Apostolidis et al., 2021; Wu et al., 2017). However, navigating through such vast content poses an exceedingly difficult challenge for users, impeding their ability to pinpoint specific points of interest within recordings (Anne Hendricks et al., 2017; Apostolidis et al., 2021). Consequently, Video Highlight Detection (HD; (Badamdorj et al., 2022; Mahasseni et al., 2017; Wei et al., 2022; Zhang et al., 2016)) and Moment Retrieval (MR; (Anne Hendricks et al., 2017; Gao et al., 2017; Liu et al., 2015; Escorcia et al., 2022)), which evaluate saliency scores of video clips and automatically identify significant moments (i.e., clips with the highest saliency scores) for user queries, respectively, have become indispensable tools in video analysis—streamlining content management, recommendation, creation, editing, and event detection processes. Given their shared objective of ranking/localizing the relevant video clips based on user queries and the commonality in their multi-modal models and data properties, recent studies using transfer learning have begun to jointly model Video Highlight Detection and Moment Retrieval (HD/MR) (Lei et al., 2021; Liu et al., 2022; Xu et al., 2023; Moon et al., 2023; Lin et al., 2023; Jang et al., 2023).

Joint HD/MR prediction requires understanding of text-video modalities and their cross-modal and cross-task synergies. Most approaches undermine either cross-task or cross-modal dynamics, limiting potential gains. Moment-DETR (Lei et al., 2021) uses concatenated pre-trained features. UMT (Liu et al., 2022) augments audio inputs but uses isolated features. QD-DETR (Moon et al., 2023) aligns text with video. MH-DETR (Xu et al., 2023) introduces a cross-modality interaction. UniVTG (Lin et al., 2023) presents multi-task learning. These methods lack cross-task interactions. TaskWeave (Yang et al., 2024) and TR-DETR (Sun et al., 2024b) address bidirectional cross-task relations, but have limitations in cross-modal dynamics. We propose **VideoLights**, a framework that leverages cross-modal and cross-task interactions through these core modules and principles:

1. **Feature Refinement and Alignment (FRA) Module**: Implements CNN-based intramodal and intermodal feature interaction and refinement, with intermodal alignment loss for text-video correspondence.

2. **Bi-Directional Cross-Modal Fusion (Bi-CMF) Network**: Employs a multi-stage hier-archical process for bidirectional text-video attention, yielding a strongly coupled query-aware clip representation.
3. **Unidirectional Joint-Task Feedback Mechanism (Uni-JFM)**: Enhances task correlation through task-specific and task-coupled losses, utilizing cosine similarity on feature vectors from HD and MR, improving cross-task learning efficiency.
4. **Adaptive Error Correction**: Incorporates hard positive and hard negative losses to adaptively penalize model errors in clip saliency prediction, fostering improved learning.
5. **Intelligent Model Pre-training**: Utilizes synthetic data generated from video corpora and language-image models to create high-quality paired text queries for model pre-training.

We perform comprehensive evaluations on widely recognized benchmarks QVHighlights (Lei et al., 2021), TVSum (Song et al., 2015), and Charades-STA (Gao et al., 2017). Results show that in both tasks, **VideoLights** achieves strong performance, outperforming all previous baselines by a significant margin (an average of 1.4% in QVHighlights, 0.7% in TVSum, and 0.3 in Charades-STA) and achieving their new state-of-the-art results. We also provide an in-depth ablation study of our model on the QVHighlights development set, visualize the qualitative examples, and analyze the effects of different synthetic pretraining corpus and the impact of feature ensembles. We will open-source our implementation accordingly.

## 2  RELATED WORK

Moment retrieval (MR) and highlight detection (HD) are related video understanding tasks. MR re-trieves video segments matching natural language queries, while HD identifies salient frames. MR approaches include two-stage (Anne Hendricks et al., 2017; Hendricks et al., 2018; Gao et al., 2017; Zeng et al., 2021; Zhang et al., 2020b; Xiao et al., 2021b) and one-stage methods (Chen et al., 2018; Liu et al., 2020; Qu et al., 2020; Ning et al., 2021; Yuan et al., 2019; Zhang et al., 2019; Zhao et al., 2021; Xiao et al., 2021a; Liu et al., 2018; Zhang et al., 2020b; 2021; Wang et al., 2021; Zhang et al., 2020a; Mun et al., 2020; Liu et al., 2021; Zeng et al., 2020)(Liu et al., 2023). Recent advance-ments in MR and HD utilize transformer-based architectures(Vaswani et al., 2017). DETR (Carion et al., 2020) simplifies predictions by eliminating anchor generation and non-maximum suppression. Moment-DETR (Lei et al., 2021) introduced the QVHighlights dataset for concurrent HD/MR, ex-celling at identifying query-relevant moments and saliency scores. UMT (Liu et al., 2022) proposed a unified multimodal architecture for MR and HD but removed the moment decoder and bipartite matching, resulting in inferior MR performance. Other approaches include TVT (Lei et al., 2020), which used subtitles, and FVMR (Gao & Xu, 2021), which improved inference speed. This paper develops a joint prediction HD/MR model focusing on cross-modal and cross-task interplays.

Cross-modal learning integrates information from different modalities, as explored in models like TERAN (Messina et al., 2021), HGSPN (Hu et al., 2019), AVS (Morgado et al., 2020), and (Badamdorj et al., 2021). Unloc (Yan et al., 2023) uses CLIP (Radford et al., 2021) for text-to-video attention in a single-stage model for multiple tasks. Our approach employs bi-directional text-video interactions with cross-task supervision. Recent works Sun et al. (2024b); Xiao et al. (2023); Moon et al. (2024) focus on feature alignment and refinement, with Yang et al. (2024) and Sun et al. (2024b) emphasizing HD-MR task interrelation. Moon et al. (2024) explores intermodality correlation for joint MR and HD.

Recent studies have explored weakly supervised pretraining with multimodal data, improving model performance (Lei et al., 2021; Xiao et al., 2023; Lin et al., 2023; Liu et al., 2022; Yan et al., 2023). Some use ASR captions as query text (Lei et al., 2021; Xiao et al., 2023; Liu et al., 2022). (Yan et al., 2023) pretrained their CLIP backend on Kinetics-700(Carreira et al., 2022) before fine-tuning. UniVTG (Lin et al., 2023) combined Ego4D (Grauman et al., 2022) and VideoCC (Nagrani et al., 2022) datasets. (Jung et al., 2022) generated two types of captions: from subtitles and visual information. In text-only contexts, (Parvez et al., 2023) demonstrated enhanced supervision by combining different encoders.

## 3  PROPOSED **VIDEOLIGHTS** MODEL

We present **VideoLights**, our joint prediction HD/MR model that enables learning from cross-modal (text vs video) and cross-task (HD vs MR) interplays. **VideoLights** features a unique

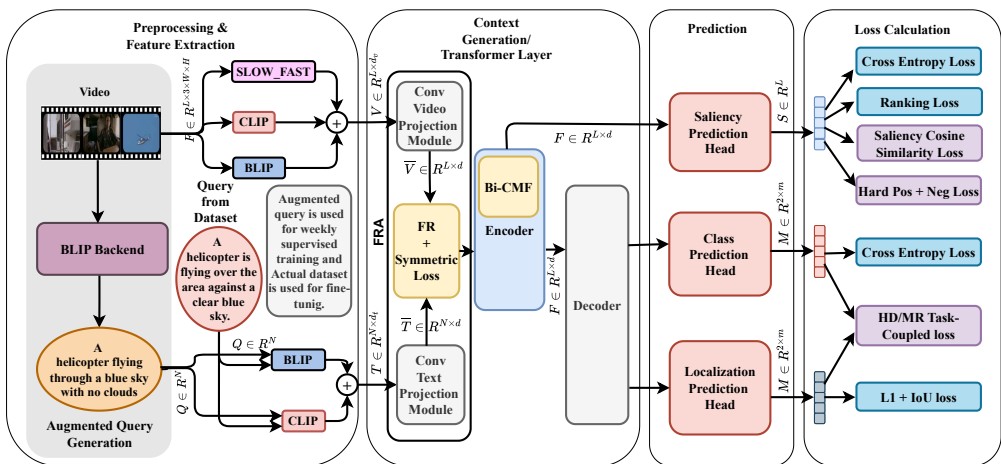

Figure 1: In `VideoLights`, FRA models the video-text cross-modal correlations from projected embeddings and passes them to Bi-CMF in the encoder. A trainable saliency vector predicts output saliency levels. Class and moment prediction heads predict logits and video moments, while saliency cosine similarity and task-coupled HD/MR losses together provide cross-task feedback *Uni-JFM*. Proposed new losses are in purple.

composite of a Bi-Directional Cross-Modal Fusion Network, a Unidirectional Join-Task Feedback module, advanced appetite loss functions, and intelligent model training. `VideoLights` pipleline is depecited in Figure 1.

### 3.1 MODEL OVERVIEW

Highlight Detection (HD) and Moment Retrieval (MR) aim to estimate the saliency of video clips and identify significant moments for a given text query. Given a video of $L$ clips, we define the video frames as $F \in \mathbb{R}^{L \times 3 \times W \times H}$, where $W$ and $H$ denote the width and height of the video, and 3 represents the number of color channels. The feature representation of the video is denoted as $V \in \mathbb{R}^{L \times d_v}$, where $d_v$ is the feature dimension extracted by a frozen video encoder. Given a text query of $N$ tokens, the representation of the text is denoted as $T \in \mathbb{R}^{N \times d_t}$, where $d_t$ is the feature dimension extracted by a frozen text encoder. With these representations and given the video and the text, our goal is twofold: for Moment Retrieval (MR), we aim to determine all the moments $M \in \mathbb{R}^{2 \times m}$, where each moment consists of a central coordinate $m_c$ and width $m_\sigma$, identifying $m$ such moments within the video. For Highlight Detection (HD), we aim to rank the saliency scores $S \in \mathbb{R}^L$ for each clip in the video to detect highlights.

**Embeddings:** We compute the initial feature sets $V$ and $T$ from multiple different VLPs as follows:

$$T = \text{clip}(Q) \oplus \text{blip}(Q) \tag{1}$$

$$V = \text{clip}(F) \oplus \text{slowfast}(F) \oplus \text{blip}(F) \tag{2}$$

Here $\oplus$ operator denotes concatenation of the features and clip, blip, and slowfast refer to frozen CLIP (Radford et al., 2021), BLIP-2 (Li et al., 2023), and Slow-Fast models (Feichtenhofer et al., 2019) respectively.

**Projection and Alignment:** When combining $V$ and $T$ for further processing, their differing hidden dimensions can make merging challenging. We address this issue by aligning the feature dimensionalities of the video and text representations using a Feed Forward Network (FFCNN) consisting of convolution layers. After this step, $V \in \mathbb{R}^{L \times d_v}$ becomes $\overline{V} \in \mathbb{R}^{L \times d}$ and $T \in \mathbb{R}^{N \times d_t}$ becomes $\overline{T} \in \mathbb{R}^{N \times d}$, where $d$ is the dimension of the hidden layer.

$$\overline{V} = \text{relu}(\text{FFCNN}(V)), \qquad \overline{T} = \text{relu}(\text{FFCNN}(T))$$

After this, we applied an intermodal feature alignment and refinement that aligned the video features with the text features. Details are discussed in Section 3.2.

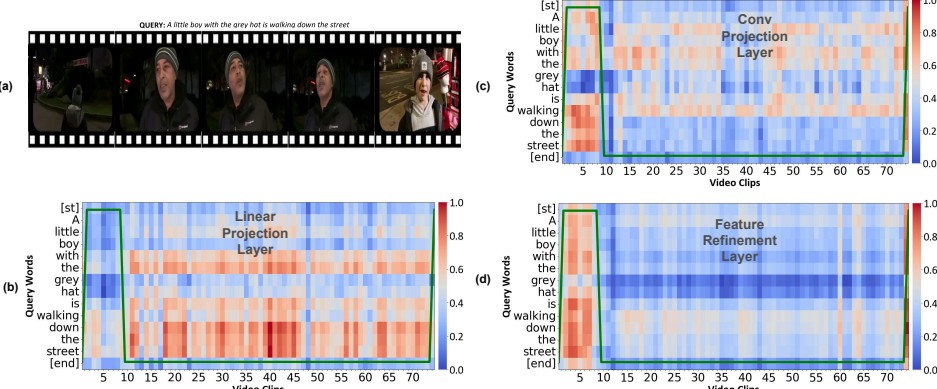

Figure 2: Here in this figure, (a) is the video, (b) and (c) are correspondence maps of query and video tokens using linear and convolution layers respectively, which show that queries are more aligned for the convolution layer, video, and text than linear projection layers. (d) is the effect of the Feature Refinement module that effectively aligns video and text tokens that match ground truth saliency levels (green line) in each heat map saliency level is shown with green line plot.

**Encoder with Cross-Modal Interaction** Both video and text representations are passed to the video-query (cross-modal) refinement module like (Sun et al., 2024a) to learn query-attended video representations and highlight relevant video tokens. Then, refined video tokens and query tokens are sent to our cross-modal interaction module *Bi-CMF* (discussed in Section 3.3). This module fuses video and text features to learn their inter-relevance and learns a strongly coupled query-injected video representation, which is then used to predict the saliency level of each clip. Then, in the multilayer encoder self, attention is applied to the output of the Bi-CMF.

**Decoder with Cross-Task Dynamics** Furthermore, the fused representation is sent to a decoder module following the work Moon et al. (2023). This module's output is used in the class prediction head and localization prediction head to predict foreground-background class and moments in video. Negative relations between irrelevant video-text queries is used to fine-tune the response, similar to what was done in (Moon et al., 2023). We propose a new learning module, unidirectional cross-task feedback network *Uni-JFM*. *Uni-JFM* takes one task HD as a reference and computes its additional losses: a task-specific (from HD) and a cross-task (from MR) losses discussed in Section 3.5.

**Adaptive Learning and Loss Functions** `VideoLights` utilizes different losses for moment retrieval and highlight identification. We utilize L1, gIoU (Union, 2019) $\mathcal{L}_{gIoU}(m, \overline{m})$, and cross-entropy $\mathcal{L}_{cls}$ objectives to perform moment retrieval like (Lei et al., 2021). Additionally, we have used margin ranking loss $\mathcal{L}_{rank}$, rank contrastive loss $\mathcal{L}_{cont}$ like (Moon et al., 2023), and entropy loss for highlight identification. Then total loss is the summation of highlight loss and moment loss. For alignment, from FRA, we used symmetric alignment loss $\mathcal{L}_{sym}$. For saliency prediction (i.e., in HD), we have introduced two adaptive hard negative loss $\mathcal{L}_{hard_{neg}}$, hard positive loss $\mathcal{L}_{hard_{pos}}$ (discussed in Section 3.4). These losses penalize errors in saliency prediction that persist with iterations.

In summary, the formulation of moment loss $\mathcal{L}_{mr}$ can be expressed as follows:

$$\mathcal{L}_{mr} = \lambda_{L1}||m - \overline{m}|| + \lambda_{gIoU}\mathcal{L}_{gIoU}(m, \overline{m}) + \lambda_{cls}\mathcal{L}_{cls} \tag{3}$$

As the additional $\mathcal{L}_{hard_{neg}}$, $\mathcal{L}_{hard_{pos}}$ as well as $\mathcal{L}_{Uni-JFM}$ losses are computed in saliency prediction, we denote the overall saliency loss as follows:

$$\mathcal{L}_{hl} = \lambda_{rank}\mathcal{L}_{rank} + \lambda_{cont}\mathcal{L}_{cont} + \mathcal{L}_{hard_{neg}} + \mathcal{L}_{hard_{pos}} + \mathcal{L}_{Uni-JFM} \tag{4}$$

Therefore, the total loss is: $\mathcal{L}_{total} = \lambda_{sal}\mathcal{L}_{hl} + \mathcal{L}_{mr} + \mathcal{L}_{sym}$ where the hyperparameters $\lambda_{sal}$ are used to achieve a balance between these losses. Below we discuss the *Bi-CMF* and *Uni-JFM* modules, Adpative $\mathcal{L}_{hard_{neg}}$, $\mathcal{L}_{hard_{pos}}$ losses, and our pretraing procedure.

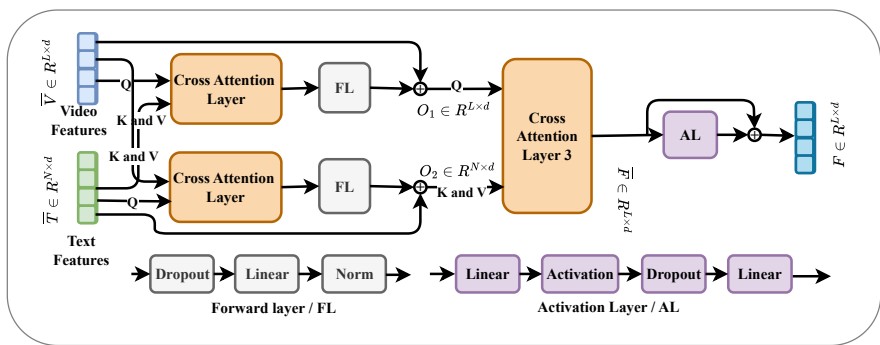

Figure 3: Bi-CMF learns query-oriented video via text2video, video2text, then text2video attentions.

## 3.2 FEATURE REFINEMENT AND ALIGNMENT NETWORK: FRA

The Feature Refinement and Alignment Network (FRA) enhances local (clip or word level) and global (video or sentence level) alignment between video and query tokens through a two-stage process. Initially, a Convolution Projection layer captures local representations, aligning video and text features while adjusting token dimensions. Subsequently, the Feature Refinement Layer achieves global alignment by computing an adjusted correspondence map, deriving sentence-level features, calculating a similarity matrix, and aggregating results. Formally, this process is represented as:

$$V_Q = \overline{V} \cdot \overline{T}^T, \qquad S = \text{pool}(\overline{T}), \qquad V_S = \overline{V} \cdot S^T,$$

$$S_v = S \cdot 1_{1 \times V \times 1}, \qquad V = \text{conv}(\overline{V} \oplus V_Q \oplus V_S \oplus S_v)$$

The FRA's effectiveness is further enhanced by a symmetric align loss adopted from Radford et al. (2021) that ensures text-to-video and video-to-text alignment, ensuring robust alignment between query and video features. The loss can be represented as:

$$\mathbf{L} = V \cdot \overline{T}^T \cdot \exp(t), \qquad \mathbf{y} = \{0, 1, 2, \ldots, n-1\},$$

$$\mathcal{L}_v = \text{CrossEntropyLoss}(\mathbf{L}, \mathbf{y}, \text{axis} = 0), \qquad \mathcal{L}_t = \text{CrossEntropyLoss}(\mathbf{L}, \mathbf{y}, \text{axis} = 1),$$

$$\mathcal{L}_{\text{sym}} = \frac{\mathcal{L}_v + \mathcal{L}_t}{2}$$

Figure 2 illustrates the FRA module's effectiveness.

## 3.3 BI-DIRECTIONAL CROSS-MODAL FUSION NETWORK: BI-CMF

To learn a strongly coupled, query-oriented video representation, we introduce our Bi-Directional Cross-Modal Fusion Network, *Bi-CMF*.

It features three multihead attention layers for cross-attention and one for self-attention. Initially, a cross-attention layer uses projected video features as queries, while text data with positional embedding serves as keys and values, identifying video tokens conditioned by textual tokens.

Similarly, another cross-attention layer is utilized to discern projected textual tokens (query) fetaures conditioned by video tokens, fused with positional embedding (keys and values), enabling the identification of textual features pertinent to the video.

Subsequently, conditioned video tokens are used as queries, while conditioned textual tokens serve as keys and values in the final cross-attention layer, yielding fused contextual information that emphasizes video tokens relevant to the query. Further refinement is achieved through a self-attention mechanism applied to this fused context, allowing for the extraction of more nuanced video context.

$$V_T = attn(\overline{V}, \overline{T}, \overline{T}), \qquad T_V = attn(\overline{T}, \overline{V}, \overline{V}), \qquad V_{attn} = attn(\overline{V}_T, \overline{T}_V, \overline{T}_V)$$

Residual connections (He et al., 2016), layer normalization (Ba et al., 2016) and dropout (Srivastava et al., 2014) mechanisms are implemented at each stage to enhance model robustness and learnable position encodings are incorporated into the input of each attention layer.

*Bi-CMF* is depicted in Figure 3 and detailed in Appendix Algorithm 2.

### 3.4 ADAPTIVE LOSS FUNCTIONS

We aim to enhance learning by identifying and rectifying persistent model errors. To achieve this, we design novel adaptive loss functions, specifically targeting hard positives and hard negatives. For the hard negative loss, we minimize the number of predictions in the negative regions where there are no relevant clips. Given the saliency score $\bar{S}_i$ and the ground truth saliency score $\mathcal{S}_i$ for non-relevant clips $i \in V_{neg}$, we define the loss, $\mathcal{L}_{hard_{neg}} = W_j \Sigma_{i \in V_{neg}} abs(\mathcal{S}_i - \bar{S}_i)$, where $W_j$ is a function of the $j$th epoch that penalizes more with a higher number of epochs. As in general, $\mathcal{S}_i$ for $i \in V_{neg}$ is zero, the loss can be defined as: $\mathcal{L}_{hard_{neg}} = W_j \Sigma_{i \in V_{neg}} abs(\bar{S}_i)$. For hard positive cases, we use Mean Square Error, and similarly, we define the loss as: $\mathcal{L}_{hard_{neg}} = W_j \Sigma_{i \in V_{pos}} MSE(\mathcal{S}_i, \bar{S}_i)$.

### 3.5 UNIDIRECTION JOINT-TASK FEEDBACK MODULE (UNI-JFM)

To leverage the cross-task synergies while jointly predicting HD/MR, we devise a unidirectional joint-task feedback mechanism that is a composite of a task-specific and a task-coupled loss. We take HD as a reference task and compute its task-specific loss $\mathcal{L}_{ts}$. To do so, we calculate the saliency cosine similarity loss from the predicted saliency level. Here for saliency score $\bar{S}$ and ground truth saliency score $\mathcal{S}$ the saliency cosine similarity loss $\mathcal{L}_{ts}$ can be defined as: $\mathcal{L}_{ts} = 1 - \frac{\bar{S}.\mathcal{S}}{\|\bar{S}\|\|\mathcal{S}\|}$. Next, for the task-coupled loss $\mathcal{L}_{tc}$, first, we use the feature vectors for MR, $M$ to calculate saliency scores $\bar{S}_{mr}$ following the MR2HD technique of (Sun et al., 2024a) using a GRU unit. Then, differently, we calculate the similarity between the ground truth saliency $\mathcal{S}$ and this calculated saliency $\bar{S}_{mr}$. This similarity score is used as the loss function $\mathcal{L}_{tc}$ , where $\mathcal{L}_{tc} = 1 - \frac{\bar{S}_{mr}.\mathcal{S}}{\|\bar{S}_{mr}\|\|\mathcal{S}\|}$. The final loss, $\mathcal{L}_{Uni-JFM} = \mathcal{L}_{ts} + \mathcal{L}_{tc}$.

### 3.6 PRETRAINING

We propose a novel multi-step methodology to enhance attention-based networks' performance by addressing limitations in ASR caption-based weakly supervised training (Lei et al., 2021; Xiao et al., 2023). Our approach segments videos into 10-second intervals, generates descriptive captions using the BLIP model for representative frames, and creates synthetic data pairs from QVHighlights and Charades-STA datasets. Saliency scores are calculated based on frame-query similarity, and the resulting caption-query pairs are used for model training. While this process may generate noisy pretrain data, the subsequent finetuning helps filter out irrelevant information, leading to improved generalization (Wu et al., 2022). Detailed data statistics and steps are provided in Appendix Table 5 and Algorithm 1.

## 4 EXPERIMENTS

**Datasets:** We evaluate **VideoLights** using three widely recognized benchmarks to ensure a comprehensive and rigorous assessment. First, the *QVHighlights* dataset (Lei et al., 2021) uniquely combines Moment and Highlight Detection tasks, providing extensive video annotations and maintaining evaluation impartiality through its online server. This dataset includes 12,562 YouTube videos and 10,310 annotations, with standardized data splits as per established works. Additionally, we use the *Charades-STA* (Gao et al., 2017) dataset for Moment Retrieval (MR) and the *TVSum* (Song et al., 2015) dataset for Highlight Detection (HD). TVSum, encompasses ten categories with five videos each. We follow the data splits in (Liu et al., 2022; Xu et al., 2023; Moon et al., 2023), that consider 80% of the dataset for training and 20% for testing. Charades-STA, features 9,848 videos and 16,128 query texts, We adopt the data splits in prior work QD-DETR (Moon et al., 2023) with 12,408 samples for training and 3,720 for testing. Our adherence to these standardized splits and the diversity of datasets underscore our commitment to a robust and fair evaluation of **VideoLights** .

**Evaluation Metrics:** We follow established evaluation metric standards from (Lei et al., 2021; Liu et al., 2022; Moon et al., 2023; Xu et al., 2023; Jang et al., 2023). For moment retrieval, we calculate Recall@1 with predetermined thresholds of 0.5 and 0.7, mean average precision (mAP) with Intersection over Union (IoU) thresholds of 0.5 and 0.75, and average mAP across multiple IoU thresholds that range from 0.50 to 0.95. The same standards are applied to the QVHighlights

Table 1: Results on QVHighlights *test* split. † represents the use of audio modality

| Method | MR | | | | | HD | |
| --- | --- | --- | --- | --- | --- | --- | --- |
| | R1 | | mAP | | | >=Very Good | |
| | @0.5 | @0.7 | @0.5 | @0.75 | Avg | mAP | HIT@1 |
| Moment-detr (Lei et al., 2021) | 52.89 | 33.02 | 54.82 | 29.4 | 30.73 | 35.69 | 55.6 |
| UMT (Liu et al., 2022) † | 56.23 | 41.18 | 53.83 | 37.01 | 36.12 | 38.18 | 59.99 |
| MH-DETR (Xu et al., 2023) | 60.05 | 42.48 | 60.75 | 38.13 | 38.38 | 38.22 | 60.51 |
| EaTR (Jang et al., 2023) | 61.36 | 45.79 | 61.86 | 41.91 | 41.74 | 37.15 | 58.65 |
| QD-DETR (Moon et al., 2023) | 62.40 | 44.98 | 63.17 | 42.05 | 41.44 | 39.13 | 63.1 |
| UVCOM (Xiao et al., 2023) | 63.55 | 47.47 | 63.37 | 42.67 | 43.18 | 39.74 | 64.20 |
| TR-DETR (Sun et al., 2024a) | 64.66 | 48.96 | 63.98 | 43.73 | 42.62 | 39.91 | 63.42 |
| TaskWeave (Yang et al., 2024) | 64.26 | 50.06 | 65.39 | **46.47** | 45.38 | 39.28 | 63.68 |
| CG-DETR (Moon et al., 2024) | 65.40 | 48.40 | 64.50 | 42.80 | 42.90 | 40.30 | 66.20 |
| UniVTG (Lin et al., 2023) | 58.86 | 40.86 | 57.60 | 35.59 | 35.47 | 38.20 | 60.96 |
| **VideoLights** | **67.51** | **51.95** | **67.13** | 45.94 | **45.72** | **41.74** | **68.09** |
| Moment-detr(pt) (Lei et al., 2021) | 59.78 | 40.33 | 60.51 | 35.36 | 36.14 | 37.43 | 60.17 |
| UMT(pt) (Liu et al., 2022) | 60.83 | 43.26 | 57.33 | 39.12 | 38.08 | 39.12 | 62.39 |
| QD-DETR (pt) (Moon et al., 2023) | 64.10 | 46.10 | 64.30 | 40.50 | 40.62 | 38.52 | 62.27 |
| UVCOM(pt) (Xiao et al., 2023) | 64.53 | 48.31 | 64.78 | 43.65 | 43.80 | 39.98 | 65.58 |
| UniVTG(pt) (Lin et al., 2023) | 65.43 | 50.06 | 64.06 | 45.02 | 43.63 | 40.54 | 66.28 |
| **VideoLights-pt** | **68.68** | **51.56** | **68.00** | **46.39** | **46.22** | **42.55** | **69.91** |

dataset. For highlight identification, our evaluations include measuring mAP and HIT@1, indicating the hit ratio for the clip with the highest score.

**Implementation details**[1]: By default, we concatenate the video fetaures, concatenating frozen *BLIP-2* (Li et al., 2023), *CLIP* (Radford et al., 2021), *Slowfast* (Feichtenhofer et al., 2019) and text features using frozen *BLIP-2* and (Li et al., 2023), *CLIP* except in TVSum. In TVSum, we follow previous wroks such as TR-DETR (Sun et al., 2024b), and use *I3D* (Carreira & Zisserman, 2017) pre-trained on Kinetics 400 (Kay et al., 2017) for visual features. We used a hidden unit size of $d = 256$, two Bi-CMF layers, three encoder layers, three decoder layers, seed value 2018, and 10-moment queries. We added a dropout rate of 0.1 for the transformer layers and 0.5 for the input projection layers (Lei et al., 2021). Loss hyperparameters were assigned as $\lambda_{L1} = 10$, $\lambda_{gIoU} = 1$, $\lambda_{cls} = 4$, $\lambda_{sal} = 1$, $\lambda_{rank} = 1$, $\lambda_{cont} = 1$, and $\Delta = 0.2$. We also initialized the model weights using the Xavier initialization (Glorot & Bengio, 2010) and tuned the model parameters with AdamW (Loshchilov & Hutter, 2019), using an initial learning rate of 1e-4 and a weight decay of 1e-4. Following (Lei et al., 2021), we trained the model for 200 epochs with a batch size of 32. For Charades-STA and TVSum, we have used a batch size of 32 and 4, respectively, with learning rates 1e-4 and 1e-3 each. See Table 6 in the appendix for details about the parameters that changed in different experiments. For all experiments, we use T4, and RTX 3050 Ti GPUs.

## 4.1 MAIN RESULTS

**Perfomance in QVHighlights:**

In Table 1, we compare the performance of various methods on the QVHighlights test split for both moment retrieval (MR) and highlight detection (HD) tasks. Our proposed framework, **VideoLights-pt** demonstrates superior performance across all metrics. Specifically, **VideoLights-pt** achieves the highest R@0.5 (68.68) and R@0.7 (51.56) for MR, and the highest mAP@0.5 (68.27) and mAP@0.75 (46.39), as well as the highest average mAP (45.22). In the HD task, **VideoLights-pt** also outperforms other methods with an mAP of 42.55 and HIT@1 of 69.91 in the pretrain fine-tuning settings. Without pretraining, **VideoLights** also achieves the best results on all but one metrics, with significant improvements over previous state-of-the-art methods: 3.23% in R1@0.5 (over CG-DETR), 3.78% in R1@0.7 (over TaskWeave), 2.66% in mAP@0.5 (over TaskWeave), 0.75% in mAP Avg (over TaskWeave), 3.57% in HD mAP, and 2.86% in HD HIT@1 (both over CG-DETR). The only metric where **VideoLights** doesn't lead is mAP@0.75, trailing TaskWeave by 1.14%. These improvements, ranging from 0.75% to 3.78%

---

[1]Codes and models are available at: TBA

Table 2: Evaluation of highlight detection methods on TVSum using Top-5 mAP. † represents the use of audio modality. ‡ indicates the use of I3D for visual feature

| Methods | VT | VU | GA | MS | PK | PR | FM | BK | BT | DS | Avg. |
|---|---|---|---|---|---|---|---|---|---|---|---|
| sLSTM (Zhang et al., 2016)‡ | 41.1 | 46.2 | 46.3 | 47.7 | 44.8 | 46.1 | 45.2 | 40.6 | 47.1 | 45.5 | 45.1 |
| SG (Mahasseni et al., 2017)‡ | 42.3 | 47.2 | 47.5 | 48.9 | 45.6 | 47.3 | 46.4 | 41.7 | 48.3 | 46.6 | 46.2 |
| LIM-S (Xiong et al., 2019)‡ | 55.9 | 42.9 | 61.2 | 54.0 | 60.3 | 47.5 | 43.2 | 66.3 | 69.1 | 62.6 | 56.3 |
| Trailer (Wang et al., 2020)‡ | 61.3 | 54.6 | 65.7 | 60.8 | 59.1 | 70.1 | 58.2 | 64.7 | 65.6 | 68.1 | 62.8 |
| SL-Module (Xu et al., 2021)‡ | 86.5 | 68.7 | 74.9 | 86.2 | 79 | 63.2 | 58.9 | 72.6 | 78.9 | 64.0 | 73.3 |
| UMT (Liu et al., 2022)†‡ | 87.5 | 81.5 | 81.5 | 81.5 | 81.4 | 87.0 | 76.0 | 86.9 | 84.4 | 79.6 | 83.1 |
| QD-DETR (Moon et al., 2023)‡ | 88.2 | 87.4 | 85.6 | 85.0 | 85.8 | 86.9 | 76.4 | 91.3 | 89.2 | 73.7 | 85.0 |
| UVCOM (Xiao et al., 2023)‡ | 87.6 | 91.6 | 91.4 | 86.7 | 86.9 | 86.9 | 76.9 | 92.3 | 87.4 | 75.6 | 86.3 |
| CG-DETR (Moon et al., 2024)‡ | 86.9 | 88.8 | **94.8** | **87.7** | 86.7 | 89.6 | 74.8 | 93.3 | 89.2 | 75.9 | 86.8 |
| TR-DETR (Sun et al., 2024a)‡ | **89.3** | **93.0** | 94.3 | 85.1 | 88.0 | 88.6 | **80.4** | 91.3 | 89.5 | **81.6** | 88.1 |
| **VideoLights** ‡ | 88.5 | 92.4 | 92.3 | 85.1 | **92.7** | 90.6 | 78.0 | 93.9 | 91.9 | 80.0 | **88.5** |
| UniVTG (Lin et al., 2023) | 83.9 | 85.1 | 89.0 | 80.1 | 84.6 | 81.4 | 70.9 | 91.7 | 73.5 | 69.3 | 81.0 |
| **VideoLights** | **90.8** | **90.6** | **89.2** | **85.0** | **88.8** | **87.6** | **73.2** | **93.0** | **87.6** | **81.8** | **86.8** |
| UniVTG (pt) (Lin et al., 2023) | **92.0** | 77.8 | 89.8 | 83.8 | 82.2 | 85.8 | 74.3 | 91.8 | **90.5** | 77.6 | 84.6 |
| **VideoLights-pt** | 88.4 | **84.7** | **91.7** | **87.0** | **90.0** | **86.4** | **77.1** | **94.0** | 88.8 | **78.7** | **86.7** |

across different metrics, show the effectiveness of our approach in both moment retrieval and highlight detection tasks.

**Perfomance in Charades-STA**

Our proposed models, **VideoLights** and **VideoLights-pt** , demonstrate competitive performance on the Charades-STA test set. Without pretraining, **VideoLights** achieves state-of-the-art results in three out of four metrics. It outperforms CG-DETR by 0.89% in R@0.5 (58.92 vs 58.40) and by 5.01% in R@0.7 (38.12 vs 36.30). **VideoLights** also improves

Table 3: Results on Charades-STA test set.

| Method | R@0.3 | R@0.5 | R@0.7 | mIoU |
|---|---|---|---|---|
| 2D-TAN (Zhang et al., 2020b) | 58.76 | 46.02 | 27.5 | 41.25 |
| VSLNet (Zhang et al., 2020a) | 60.30 | 42.69 | 24.14 | 41.58 |
| Moment-detr (Lei et al., 2021) | 65.83 | 52.07 | 30.59 | 45.54 |
| QD-DETR (Moon et al., 2023) | - | 57.31 | 32.55 | - |
| TR-DETR (Sun et al., 2024a) | - | 57.61 | 33.52 | - |
| UniVTG (Lin et al., 2023) | **70.81** | 58.01 | 35.65 | 50.10 |
| CG-DETR (Moon et al., 2024) | 70.40 | 58.40 | 36.30 | 50.10 |
| **VideoLights** | 70.73 | **58.92** | **38.12** | **50.77** |
| UniVTG (pt) (Lin et al., 2023) | **72.63** | 60.19 | 38.55 | **52.17** |
| **VideoLights-pt** | 72.28 | **60.54** | **38.95** | 51.73 |

upon UniVTG's mIoU by 1.34% (50.77 vs 50.10). For R@0.3, **VideoLights** (70.73) closely trails UniVTG (70.81) by a marginal 0.11%. In the pretraining setting, **VideoLights-pt** shows mixed results compared to UniVTG (pt). It surpasses UniVTG (pt) by 0.58% in R@0.5 (60.54 vs 60.19) and by 1.04% in R@0.7 (38.95 vs 38.55). However, **VideoLights-pt** falls slightly behind in R@0.3 by 0.48% (72.28 vs 72.63) and in mIoU by 0.84% (51.73 vs 52.17). These results highlight the effectiveness of our approach, particularly in improving performance on stricter evaluation criteria (R@0.5 and R@0.7) in both pretraining and non-pretraining scenarios.

**Perfomance in TVSum:** Our proposed model **VideoLights** demonstrates competitive performance across various domains in the TVSum dataset, as shown in Table 2. **VideoLights** achieves state-of-the-art results in 4 out of 10 domains and in the overall average. Specifically, it outperforms previous methods in PK (92.7% vs TR-DETR's 88.0%, a 5.34% improvement), PR (90.6% vs CG-DETR's 89.6%, a 1.12% gain), BK (93.9% vs CG-DETR's 93.3%, a 0.64% increase), and BT (91.9% vs TR-DETR's 89.5%, a 2.68% improvement). In the remaining domains, **VideoLights** shows competitive performance, closely trailing the best results: VT (88.5% vs TR-DETR's 89.3%, -0.89%), VU (92.4% vs TR-DETR's 93.0%, -0.65%), GA (92.3% vs CG-DETR's 94.8%, -2.64%), MS (85.1%, tied with TR-DETR), FM (78.0% vs TR-DETR's 80.4%, -2.99%), and DS (80.0% vs TR-DETR's 81.6%, -1.96%). Notably, **VideoLights** achieves the highest overall average performance of 88.5%, surpassing TR-DETR's 88.1% by 0.45%. These results highlight the effectiveness of **VideoLights** across diverse video domains in highlight detection tasks. When compared with UniVTG without pretraining, case **VideoLights** outperforms in all domains, and with pertaining, case **VideoLights-pt** outperforms in all domains except VT and BT.

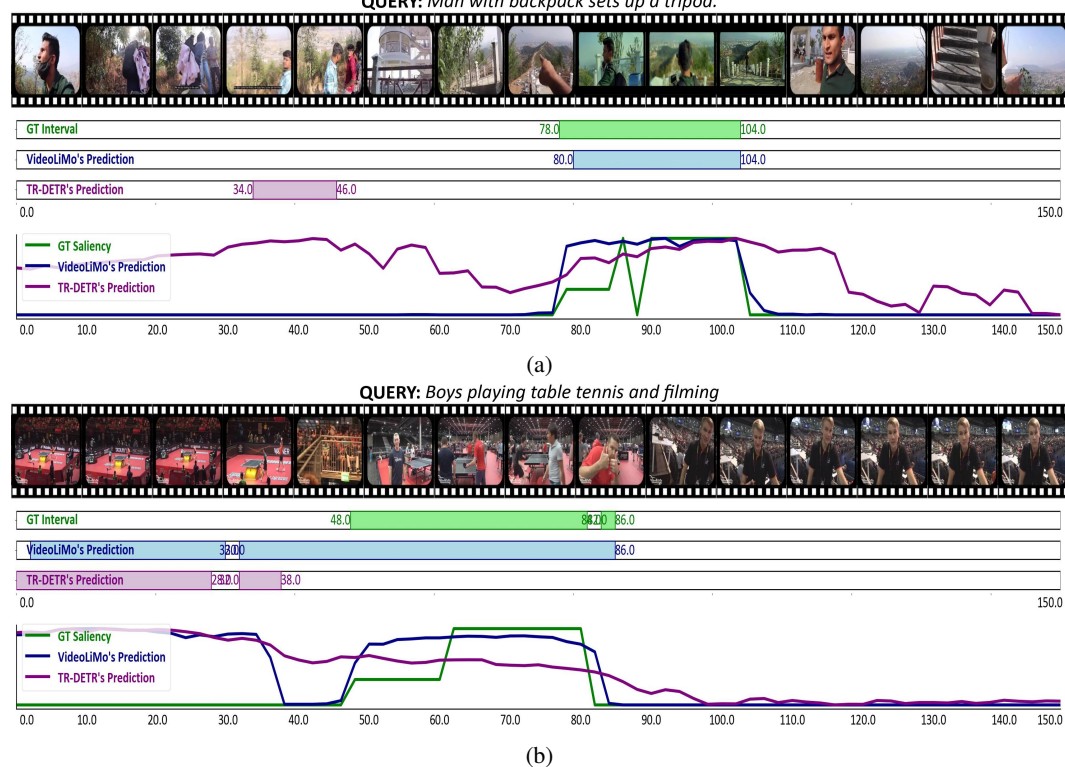

(a)

(b)

Figure 4: Qualitative results. **(a)** demonstrates `VideoLights` outperformed TR-DETR (Sun et al., 2024b) in both MR and HD. **(b)** Both `VideoLights` and TR-DETR performed below the ground truth, but upon closer examination, it is evident that incorrectly predicted clips are still related to the given query.

In summary, `VideoLights` not only matches but often exceeds the performance of other cutting-edge methods, demonstrating its effectiveness in joint video highlight detection & moment retrieval.

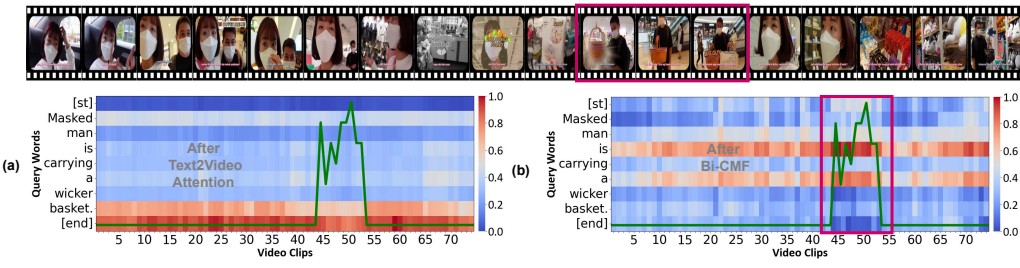

Figure 5: (a) and (b) show video-query correspondence maps: (a) after text-to-video (t2v) attention and (b) after the Bi-CMF layer. The green line represents the ground truth saliency scores. Bi-CMF attends to the correct video region better than t2v (highlighted in the magenta box). The word 'Is' asserts that 'a' refers to one basket, unlike 'is not'.

## 4.2 ABLATION STUDIES

To comprehend module impacts, we present our model ablation on QVHighlights *val* split in Table 4.

**Effect of FRA:** From Table 4 comparing rows 2 and 5, the addition of the FRA module while keeping Bi-CMF disabled results in an average performance gain of 9.24% across all metrics. Also, Figure 2 shows the qualitative efficacy of this module.

**Effect of Bi-CMF:** The rows 2 and 4 of Table 4 demonstrate the effectiveness of our *Bi-CMF* module, showing an average performance gain of 4.41% across all metrics, with the most significant

Table 4: Ablation study on QVHighlights val split. fra stands for FRA module, bi stands for Bi-CMF module, bf stans for Blip features, pt stands for pre-train on the synthetic dataset using Blip Backend, hl stands for adaptive hard positive and negative loss, tcl stands for task coupled loss, and scsl stands for saliency cosine similarity loss. The effect of different pretraining data is in the bottom block without any new losses.

| | Modules | | | | Losses | | | MR | | | | | HD | |
| | | | | | | | | R1 | | mAP | | | >=Very Good | |
| sl. | fra | bi | bf | pt | hl | tcl | scsl | @0.5 | @0.7 | @0.5 | @0.75 | Avg | mAP | HIT@1 |
|---|---|---|---|---|---|---|---|---|---|---|---|---|---|---|
| 1. | ✗ | ✗ | ✗ | ✗ | ✓ | ✓ | ✓ | 60.77 | 45.74 | 61.24 | 41.32 | 40.71 | 37.91 | 58.71 |
| 2. | ✗ | ✗ | ✓ | ✗ | ✓ | ✓ | ✓ | 62.13 | 49.03 | 62.92 | 44.20 | 44.04 | 39.67 | 63.87 |
| 3. | ✓ | ✓ | ✗ | ✗ | ✓ | ✓ | ✓ | 63.16 | 48.00 | 63.25 | 43.96 | 43.39 | 39.64 | 63.03 |
| 4. | ✗ | ✓ | ✓ | ✗ | ✓ | ✓ | ✓ | 65.42 | 52.84 | 64.89 | 46.67 | 45.69 | 40.75 | 65.55 |
| 5. | ✓ | ✗ | ✓ | ✗ | ✓ | ✓ | ✓ | 70.45 | 54.26 | 68.88 | 47.61 | 47.50 | 42.47 | 69.29 |
| 6. | ✓ | ✓ | ✓ | ✗ | ✓ | ✓ | ✓ | 70.26 | 54.84 | 68.90 | 48.77 | 47.87 | 42.19 | 69.48 |
| 7. | ✓ | ✓ | ✓ | ✗ | ✗ | ✗ | ✗ | 63.81 | 48.00 | 64.39 | 43.64 | 43.02 | 39.12 | 63.68 |
| 8. | ✓ | ✓ | ✓ | ✗ | ✗ | ✓ | ✗ | 68.58 | 53.42 | 67.96 | 47.75 | 47.7 | 42.38 | 68.71 |
| 9. | ✓ | ✓ | ✓ | ✗ | ✗ | ✓ | ✗ | 69.10 | 53.87 | 68.56 | 47.73 | 48.02 | 41.69 | 67.94 |
| 10. | ✓ | ✓ | ✓ | ✗ | ✗ | ✗ | ✓ | 68.90 | 54.26 | 68.68 | 48.94 | 47.88 | 42.65 | 70.26 |
| 11. | ✓ | ✓ | ✓ | ✓ | ✓ | ✓ | ✓ | **71.74** | **56.77** | **69.35** | **50.09** | **49.10** | **43.38** | **71.29** |
| No Pretraining | | | | | | | | 63.29 | 44.52 | **63.49** | 39.83 | 39.96 | 38.37 | 62.84 |
| ASR Pretraining (Lei et al., 2021) | | | | | | | | 61.42 | 44.97 | 62.25 | 40.37 | 40.08 | 38.28 | 61.16 |
| Our BLIP Pretraining | | | | | | | | **63.23** | **46.00** | 62.67 | **41.32** | **40.71** | **39.89** | **63.87** |

improvement in R@0.7 (7.77%). A qualitative analysis through feature heatmap visualization in Figure 5 reveals that *Bi-CMF* achieves a more sparse spectrum density compared to both baseline (no cross-modal) and uni-directional (text-to-video) approaches like QD-DETR, indicating better query relevance differentiation.

**Effect of new loss functions:** Row 7-10 in Table 4 in the top block signify the gains using our proposed loss functions. All of the new losses show significant improvements in both tasks individually and in combination, which enhances tasks to a great extent. Here, we see that hl and scsl contribute towards HD, and tcl contributes to MR tasks.

**Effect of Blip-2 features and Pretraining:** As shown especially the difference between the 6th row and the 11th row in the upper block, pre-training also helps improve performance. Usage of BLIP-2 features along with the standard CLIP, and SlowFast also brings about improvements. The bottom block shows the results with different pretraining corpus that poses the effectiveness of pretraining.

# 5 LIMITATION AND CONCLUSION

**Conclusion** In this paper, we propose a novel joint prediction model for highlight detection and moment retrieval, `VideoLights`. It features a feature refinement and alignment module, a bi-directional cross-modal and uni-directional cross-task feedback mechanism. Our custom cross-modal interaction module enhances the ability to understand intermodal relationships between text and video, resulting in superior content retrieval and highlight identification performance. Our experiments on the QVHighlights and TVSum datasets have shown that our approach outperforms current techniques and has fewer learning parameters, indicating efficiency and scalability. Our contributions set the stage for future research in video content analysis. demonstrating the potential of integrating advanced language and vision models to tackle real-world challenges in multimedia content processing.

**Limitation** Our proposal for weakly supervised training utilizing vision-language pretraining models simplifies the training process but may still be prone to biases or inaccuracies in caption generation. At the same time, our dependency on pretraining models for caption generation and feature extraction can lead to computational overhead and reliance on external resources, thus potentially limiting the scalability of our approach. Moreover, the performance of our Bi-CMF module is heavily reliant on the quality of input features and the effectiveness of attention mechanisms, both of which can vary depending on the complexity and diversity of the video content. To fully unlock the potential of our proposed approach in real-world applications, it is crucial to address these limitations through further research and refinement.

## 6 REPRODUCIBILITY STATEMENT

To ensure the reproducibility of our experimental results, we provide comprehensive details of our implementation. The core hyperparameters and environmental settings used across all experiments are thoroughly documented in Section 4. For specific experiments that required parameter tuning, we present a detailed breakdown in Table 6, which includes the optimal hyperparameter configurations for each dataset and evaluation scenario. This includes learning rates, batch sizes, and model-specific parameters that were determined through empirical validation. The complete source code, including pre-processing scripts, model architectures, training pipelines, and evaluation protocols, along with detailed instructions for environment setup and data preparation, is available in the supplementary materials. We shall provide model checkpoints and experiment logs to ensure reproducibility.

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

# A APPENDIX

## A.1 DATASET STATISTICS

Table 5 provides a comparison of three datasets utilized in a study, describing the different attributes of each. The QVHighlights dataset includes vlog and news content, with 10,300 annotations and 12,500 videos. It supports tasks such as Moment Retrieval (MR) and Highlight Detection (HD) and has been utilized in pre-training. We have generated 187682 synthetic data from videos of this dataset using the approach described in Algorithm 1. The Charades-STA dataset, which focuses on activity-related content, comprises 16,100 annotations and 6,700 videos, specifically used for Moment Retrieval and has also been employed in pre-training. We have generated 23,193 synthetic data samples from this dataset. Lastly, the TVSum dataset, based on web content, is notably smaller, with 50 annotations and 50 videos, used exclusively for Highlight Detection. It has 10 domains, VT, VU, GA, MS, PK, PR, FM, BK, BT, and DS each containing 5 videos. Unlike the other datasets, it has not been used in pre-training and does not include synthetic data.

Table 5: Comparison of datasets used in this study.

| Dataset | Domain | Annotations | Videos | Task | Used in pt | Synthetic data |
|---------|--------|-------------|--------|------|------------|----------------|
| QVHighlights | Vlog / News | 10.3K | 12.5K | MR, HD | ✓ | 187682 |
| Charades-STA | Activity | 16.1K | 6.7K | MR | ✓ | 23193 |
| TVSum | Web | 50 | 50 | HD | | |

---

**Algorithm 1** Synthetic data generation process

---

1: Segment videos into 10-second intervals, each representing a discrete moment within the video content.
2: For each 10-second interval, select a representative frame and use the BLIP model to generate a descriptive caption for that frame.
3: Use the generated caption as a query, encapsulating the essence of the selected frame.
4: Match the generated query-captions with video frames within each 10-second interval using cosine similarity to find the similarity score, which serves as the saliency level for highlight detection.
5: Train the model using the generated caption-query pair, considering the entire 10-second interval as a moment for training purposes.

---

## A.2 ADDITIONAL ABLATION ON BI-CMF

Our research findings indicate that integrating the Bi-CMF module into our model significantly improves performance in MR and HD tasks compared to the model without this module. In addition, we conducted further ablation studies to evaluate the impact of different Bi-CMF layer counts on the model. The results, outlined in Table 8, show that while one and two layers demonstrate similar performance in both MR and HD metrics, the introduction of three layers enhances MR performance but decreases performance in HD tasks. Furthermore, as the number of layers increases, performance on both tasks across all metrics decreases. The impact on MR performance, particularly in MR-full-mAP, is illustrated in Figure 6.

We also performed additional experiments to assess the effectiveness of bi-CMF compared to unidirectional cross-attention. In this experiment, we replaced our bi-CMF with a unidirectional cross-attention module while keeping all other parameters constant. The results are presented in Table 9. We observed that across all metrics in the MR task, Bi-CMF demonstrated a notable improvement over unidirectional cross-attention.

## A.3 SOCIETAL IMPACT

The research explores significant societal implications of the advancements in Video Highlight Detection and Moment Retrieval (HD/MR). With the exponential growth of online video content, these

---

**Algorithm 2** Bi-Directional Cross-Modal Fusion Network

---

1: **Input:** Video embeddings $\overline{V}$, Text embeddings $\overline{T}$
2: **Output:** Fused contextual information $F$
3: Initialize $F$ as empty tensor
4: # Apply cross-attention between $\overline{V}$ and $\overline{T}$ to obtain video tokens conditioned by text tokens:
5: Query $= \overline{V} \cdot W_q$
6: Key $= \overline{T} \cdot W_k + $ PositionalEmbedding
7: Value $= \overline{T} \cdot W_v + $ PositionalEmbedding
8: $O_1 = \text{Softmax}(\frac{\text{Query} \cdot \text{Key}^\top}{\sqrt{d}}) \cdot$ Value
9: $O_1 = \overline{V} + norm(linear(dropout(O_1)))$
10: # Apply cross-attention between $\overline{T}$ and $\overline{V}$ to obtain text tokens conditioned by video tokens:
11: Query $= \overline{T} \cdot W_q$
12: Key $= \overline{V} \cdot W_k + $ PositionalEmbedding
13: Value $= \overline{V} \cdot W_v + $ PositionalEmbedding
14: $O_2 = \text{Softmax}(\frac{\text{Query} \cdot \text{Key}^\top}{\sqrt{d}}) \cdot$ Value
15: $O_2 = \overline{T} + norm(linear(dropout(O_2)))$
16: # Cross-attention to $O_1$ and $O_2$ to obtain fused representation:
17: Query $= O_1 \cdot W_q$
18: Key $= O_2 \cdot W_k$
19: Value $= O_2 \cdot W_v$
20: $O_3 = \text{Softmax}(\frac{\text{Query} \cdot \text{Key}^\top}{\sqrt{d}}) \cdot$ Value
21: # Apply self-attention with layer normalization to obtain fine grained representation:
22: Query $= O_3 \cdot W_q$
23: Key $= O_3 \cdot W_k$
24: Value $= O_3 \cdot W_v$
25: $\overline{F} = \text{Softmax}(\frac{\text{Query} \cdot \text{Key}^\top}{\sqrt{d}}) \cdot$ Value
26: $\overline{F} = O_3 + dropout(\overline{F})$
27: $\overline{F}_d = dropout(activation(linear(\overline{F})))$
28: $F = norm(linear(\overline{F}_d))$
29: **return** $F$

---

Table 6: Experiment-specific hyperparameters. Visual features: I3D, SlowFast (SF), CLIP (C), and BLIP-2 (Blip). VF: visual features, TF: text features. Coefficients: symmetric alignment loss (al_coef), task coupled loss (tcl_coef), hard positive/negative loss (hl_coef), and cosine similarity loss (scsl_coef).

| Dataset | Exp | VF | TF | Epoch | lr | Bs | al_coef | tcl_coef | hl_coef | scsl_coef |
|---|---|---|---|---|---|---|---|---|---|---|
| QVHighlights | Without pt | SF+C+Blip | C+Blip | 200 | 1E-04 | 32 | 0.1 | 1 | 10 | 1 |
|  | Finetune | SF+C+Blip | C+Blip | 200 | 1E-04 | 32 | 0.8 | 1 | 10 | 1 |
| Charades-STA | Without pt | SF+C+Blip | C+Blip | 100 | 1E-04 | 32 | 0.1 | 1 | 10 | 1 |
|  | Finetune | SF+C+Blip | C+Blip | 100 | 1E-04 | 32 | 0.8 | 1 | 10 | 1 |
| TVSum | I3D | I3D+Blip | C+Blip | 2000 | 1E-03 | 4 | 0.8 | 0 | T 7 | T 7 |
|  | SF+C+Blip | SF+C+Blip | C+Blip | 2000 | 1E-03 | 4 | 0.8 | 0 | T 7 | T 7 |
|  | Finetune (SF+C+Blip) | SF+C+Blip | C+Blip | 2000 | 1E-03 | 4 | T 7 | 0 | T 7 | T 7 |

technologies have the potential to greatly improve user experiences by facilitating easy navigation and retrieval of pertinent information within videos. This could result in more efficient consumption of educational material, greater accessibility for individuals with limited time or attention spans, and better organization of news and entertainment media. However, the societal impact extends beyond the mere convenience. These tools could also be used to automate video summarization for surveillance footage or body camera recordings, raising privacy concerns and ethical questions regarding AI-driven video analysis. While technology offers numerous benefits, it is imperative to carefully consider potential misuse, such as creating deceptive video summaries or perpetuating algorithmic biases in video highlight generation systems. Therefore, continuing ethical debates and responsible

Table 7: Value of hl_coef, scsl_coef and al_coef in different experiments on the TVSum dataset

| Exp | Coef Name | VT | VU | GA | MS | PK | PR | FM | BK | BT | DS |
|---|---|---|---|---|---|---|---|---|---|---|---|
| I3D | hl_coef | 10 | 1 | 1 | 1 | 10 | 10 | 10 | 1 | 10 | 10 |
| | scsl_coef | 1 | 1 | 10 | 5 | 5 | 10 | 5 | 1 | 1 | 10 |
| SF+C+Blip | hl_coef | 10 | 10 | 10 | 10 | 10 | 10 | 10 | 10 | 10 | 10 |
| | scsl_coef | 10 | 5 | 10 | 1 | 5 | 10 | 1 | 10 | 10 | 10 |
| Finetune (SF+C+Blip) | hl_coef | 1 | 1 | 5 | 1 | 5 | 1 | 5 | 1 | 5 | 10 |
| | scsl_coef | 5 | 1 | 1 | 1 | 10 | 10 | 10 | 5 | 10 | 5 |
| | al_coef | 0.8 | 0.8 | 0.1 | 0.8 | 0.1 | 0.8 | 0.1 | 0.8 | 0.1 | 0.8 |

Table 8: Experiment using different Bi-CMF layer counts on QVHighlights *val* split.

| Bi-CMF layer count | MR | | | | | HD | |
|---|---|---|---|---|---|---|---|
| | R1 | | mAP | | | >=Very Good | |
| | @0.5 | @0.7 | @0.5 | @0.75 | Avg | mAP | HIT@1 |
| 0 | 65.55 | 49.74 | 64.50 | 44.51 | 43.86 | 40.87 | 66.9 |
| 1 | 68.84 | 53.16 | 67.29 | 46.08 | 45.98 | 42.31 | 69.35 |
| 2 | 68.84 | 53.10 | 67.41 | 46.59 | 45.88 | **42.20** | **69.61** |
| 3 | **69.16** | **52.71** | **68.29** | **47.27** | **47.27** | 42.13 | 67.74 |
| 4 | 67.16 | 52.58 | 66.95 | 47.21 | 46.55 | 41.47 | 67.35 |
| 5 | 68.00 | 52.58 | 66.86 | 46.58 | 46.11 | 41.12 | 67.94 |

Table 9: Experiment using different Bi-CMF layer counts on QVHighlights *val* split.

| Method | MR | | | | | HD | |
|---|---|---|---|---|---|---|---|
| | R1 | | mAP | | | >=Very Good | |
| | @0.5 | @0.7 | @0.5 | @0.75 | Avg | mAP | HIT@1 |
| UniDirectional Attention | 67.61 | 50.65 | 67.06 | 45.46 | 45.42 | **42.59** | **69.61** |
| Bi-CMF | **68.84** | **53.16** | **67.29** | **46.08** | **45.98** | 42.31 | 69.35 |

Table 10: Effect of hl_coef and scl_coef on TVSum result on I3D visual features

| hl_coef | scl_coef | VT | VU | GA | MS | PK | PR | FM | BK | BT | DS | Avg. |
|---|---|---|---|---|---|---|---|---|---|---|---|---|
| 10 | 1 | **88.45** | 85.32 | 83.43 | 80.85 | 84.18 | 87.13 | 77.1 | 92.36 | **91.92** | 76.88 | 84.76 |
| | 5 | 84.31 | 71.32 | 91.93 | **85.13** | **92.67** | 84.08 | **78.01** | 91.59 | 90.46 | 77.5 | 84.70 |
| | 10 | 87.29 | 75.32 | 82.68 | 80.67 | 87.43 | **90.58** | 72.55 | 91.68 | 86.85 | **79.99** | 83.50 |
| 1 | 1 | 87.29 | **92.43** | 85.63 | 81.76 | 79.87 | 85.55 | 63.81 | **93.96** | 85.72 | 63.92 | 81.99 |
| | 5 | 83.77 | 75.63 | 88.97 | 79.71 | 80.65 | 87.56 | 72.55 | 90.79 | 88.83 | 77.1 | 82.56 |
| | 10 | 87.36 | 75.479 | **92.29** | 85.02 | 84.56 | 87.69 | 71.73 | 91.25 | 87.08 | 77.93 | 84.04 |
| Max | | 88.45 | 92.43 | 92.29 | 85.13 | 92.67 | 90.58 | 78.01 | 93.96 | 91.92 | 79.99 | 88.54 |

development practices will be indispensable as these technologies progress and become integrated into various aspects of society.

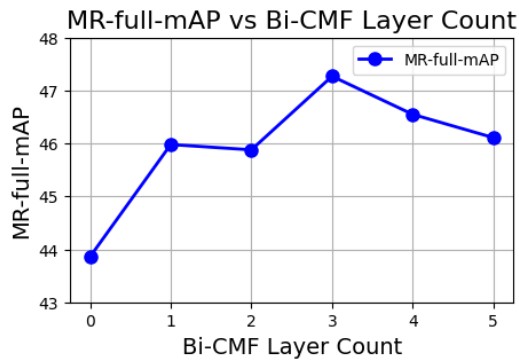

Figure 6: Bi-CMF layer count VS MR-full-mAP plot

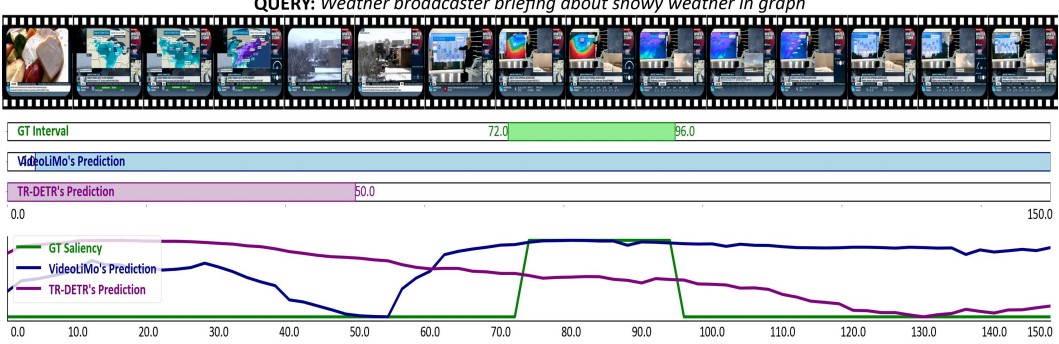

Figure 7: Qualitative results. In case there is little change in consecutive frames, our model failed to detect moments properly.

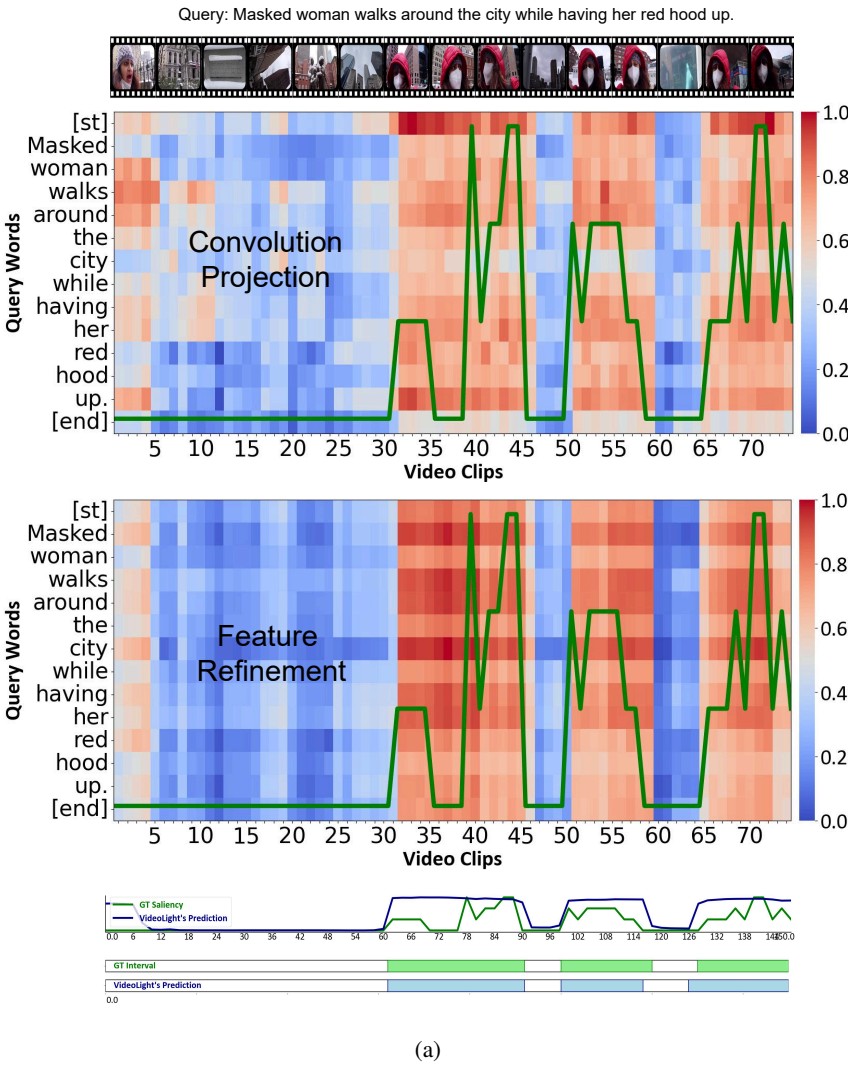

(a)

Figure 8: Qualitative results. demonstrates when FRA aligned video and query better **VideoLights** was able to predict better. Here the green line plot and bar are respectively ground truth HD and MR results, and blue one is **VideoLights** prediction.

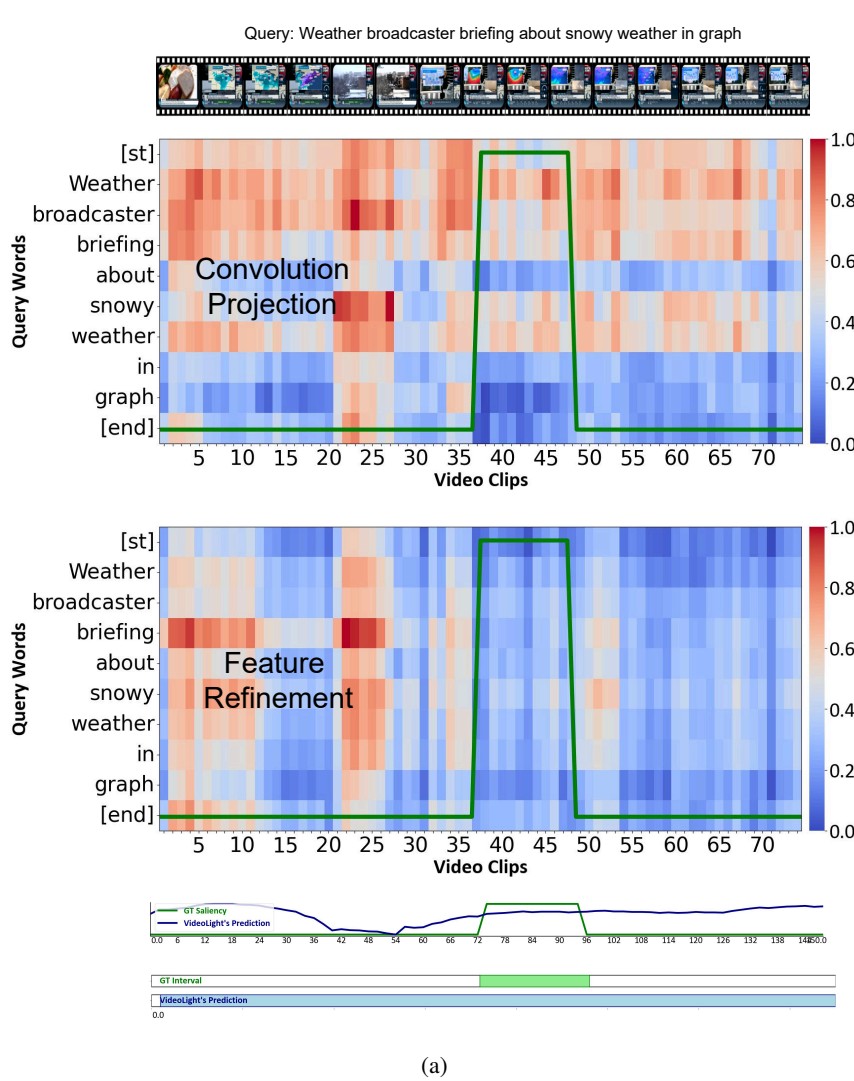

(a)

Figure 9: Qualitative results. When FRA failed to align word and video, **VideoLights** failed to predict better MR and HD. Here the green line plot and bar are respectively ground truth HD and MR results, and blue one is **VideoLights** prediction.

