# OpenReview forum: "VideoLights: A Cross-Modal Cross-Task Transformer Model for Joint Video Highlight Detection and Moment Retrieval"
_ICLR.cc/2025/Conference — ICLR 2025 Conference Withdrawn Submission_

### Official Review · Reviewer_u5Zv · 2024-10-27

**Soundness:** 2
**Presentation:** 1
**Contribution:** 1
**Rating:** 3
**Confidence:** 5

**Summary:**

This paper jointly tackles the problem of video moment retrieval and highlight detection (MR/HD). The authors claimed that existing works in such a direction overlook the problem of 'cross-task dynamics' and 'video-text alignment'. Therefore, they proposed VideoLights, a new MR/HD method with three contributions: 1) convolutional projection and feature refinement, 2) bi-directional cross-modal fusion, and 3) uni-directional join-task feedback mechanism. Experiments on public datasets demonstrate the effectiveness of the propose method.

**Strengths:**

1. Overall, the flow of the paper is easy to follow. This work focuses on an existing well-defined setting of moment retrieval and highlight detection.
2. The experimental results are good. The proposed method obtained near state-of-the-art performance.

**Weaknesses:**

My major concerns on this work focus on the motivation, novelty, and experiments.

1. The motivation of this work is unclear. In the very short introduction, the authors did not provide in-depth analysis on 'why' proposing such designs but only focused on introducing 'how' the proposed complicated modules work. The claimed limitations of existing works (lacking cross-task dynamics and video-text alignment) were not explained nor mentioned in the experiments. It's better to have more discussions on the motivations rather than simply providing detailed module designs.
2. Given the unclear motivation, the novelty of the proposed method is limited as well. Combining multiple visual encoders (CLIIP + SlowFast + BLIP) and introducing cross-modal/uni-modal/bi-directional/uni-directional connections without strong justifications on the reasons cannot convince the readers how these modules can provide better MR and HD results. Besides, similar designs have already been widely discussed/used in existing works.
3. For the experiments, some representative methods [1,2,3,4] were not mentioned or compared. It would be better to have more discussions on these related works, either by including these methods in their experiments or by providing a detailed discussion of how their approach compares to these methods theoretically.
4. Detailed analysis focusing on the efficiency (parameter and time) is needed.

[1] Unloc: A unified framework for video localization tasks. ICCV 2023
[2] Knowing Where to Focus: Event-aware Transformer for Video Grounding. ICCV 2023
[3] MomentDiff: Generative Video Moment Retrieval from Random to Real. arXiv 2023
[4] R2-tuning: Efficient image-to-video transfer learning for video temporal grounding. ECCV 2024

**Questions:**

In table 2, why the performance of 'VideoLights-pt' is worse than 'VideoLights'?

---

### Official Review · Reviewer_hGXz · 2024-10-29

**Soundness:** 3
**Presentation:** 3
**Contribution:** 2
**Rating:** 3
**Confidence:** 2

**Summary:**

This manuscript introduces, Videolights, designed for moment retrieval and highlight detection. It consists of the three modules. First, a convolutional projection and feature refinement module aligns video with query text to enhance their compatibility. Second, a bidirectional cross-modal fusion network enables query-aware feature representation, ensuring both modalities interact effectively. Lastly, a uni-directional joint-task feedback mechanism is proposed to optimize performance. Experimental results confirm the effectiveness of this approach.

**Strengths:**

__S1.__ The overall method is easy to understand.

__S2.__ The proposed method demonstrates performance contributions across three datasets (QVHighlights, TVSum, and Charades-STA).

**Weaknesses:**

__W1.__ Although performance improvements have been observed across various datasets, it seems that the fusion of three visual encoders (Slowfast, CLIP, BLIP) also has a significant impact. Additional experiments are needed to investigate the performance variations based on the types of visual encoders.

__W2.__ When I see Figure 4, the authors compare with TR-DETR. Is there a specific reason why they chose TR-DETR for the QVHighlights dataset instead of comparing it with the state-of-the-art methods (e.g., CG-DETR and UniVTG)?

__W3.__ The technical novelty is limited. The concept of bidirectional cross-modal learning has been introduced in several works (e.g., [Ref_1]).

__W4.__ Overall, it seems to represent a fusion of existing technical components, such as bidirectional cross-modal learning and uni-directional joint-task feedback mechanisms.

__W5.__ One interesting aspect is 'query generation,' as the quality of query generation could significantly impact performance. However, there is a lack of experiments in this method, and an explanation for why BLIP was chosen is needed. Additionally, what would happen if other models were aopted?

[Ref_1] W. Wu et al., "Bidirectional Cross-Modal Knowledge Exploration for Video Recognition with Pre-trained Vision-Language Models," in CVPR 2023

**Questions:**

In addition to the weakness,

__Q1.__ What is the technical novelty of the proposed method?

__Q2.__ The authors mention in the introduction that existing methods lack cross-modal dynamics. Is there any evidence to support this assertion, aside from performance metrics?

__Q3.__ There is no explanation regarding the motivation for the proposed method in the introduction. A convincing rationale for this is needed.

---

### Official Review · Reviewer_57Dj · 2024-11-02

**Soundness:** 1
**Presentation:** 2
**Contribution:** 2
**Rating:** 3
**Confidence:** 5

**Summary:**

This paper presents the VideoLight framework for Video Highlight Detection and Moment Retrieval (HD/MR) tasks, emphasizing cross-task dynamics and video-text alignment. Firstly, the authors aim to enhance video-text alignment through the Feature Refinement and Alignment (FRA) Module. Additionally, they propose the Bi-Directional Cross-Modal Fusion (Bi-CMF) Network, moving beyond simple cross-attention-based encoding of text and video to learn a strongly coupled, query-oriented video representation. Furthermore, they introduce adaptive loss, coupled loss, and saliency cosine similarity loss to enhance cross-task synergies and address persistent model errors.

**Strengths:**

1. The paper focuses on the emerging topic of cross-task dynamics in HD/MR tasks, proposing novel methods and losses to address this challenge.

2. The authors provide evaluations across diverse datasets and conduct ablation studies for the numerous proposed modules and losses.

3. They improve performance by using BLIP features, which have not been widely used in MR tasks, and also apply a pre-trained approach.

**Weaknesses:**

1. This paper adds several modules, but it lacks an in-depth analysis of each module’s specific impact beyond just improving HD/MR performance. Futhermore, a comparison of rows 5 and 6 in the ablation table shows that the Bi-CMF module has less effect on performance.

2. Unlike previous methods that only used video/text features (CLIP, SlowFast or  I3D) for fair comparison, the authors evaluate their model by additionally incorporating BLIP2 features. This introduces an unfair comparison.

3. The paper appears to be in the process of refinement, as there are inconsistencies in terminology. For example, while the proposed framework is named "VideoLight," Figures 4 and 7 label it as "VideoLimo." Additionally, terminology varies throughout the paper, which can lead to some confusion.

**Questions:**

My major concerns are listed in the Weaknesses section, along with additional comments and further questions below.

1. As mentioned in Weakness 2, for a fair comparison, it would be preferable to either add the BLIP feature to existing methods or evaluate the proposed method without the BLIP feature. The ablation study in the paper shows that simply using the BLIP feature significantly improve MR/HD performance.

2. The Feature Refinement and Alignment (FRA) module appears to have the greatest impact on improving MR performance among the proposed modules, according to the ablation study. To demonstrate the effectiveness of FRA, the authors provide qualitative text-video token correspondence maps in Figures 2 and 8. However, beyond these qualitative results from specific samples, they should also verify quantitatively across the entire evaluation dataset whether the correspondence between text tokens and video clips relevant to moments has increased.

3. The losses proposed by the authors could be applied to other existing methods, and experiments on this would be included. The performance improvements appear to be the result of technical adjustments, as the coefficients for the three additional losses vary across datasets(in the appendix).

---

### Official Review · Reviewer_wpWM · 2024-11-05

**Soundness:** 3
**Presentation:** 2
**Contribution:** 3
**Rating:** 5
**Confidence:** 4

**Summary:**

This work studies the highlight detection and moment retrieval problem, and introduces multiple modules/mechanisms to address the issue of considering cross-task dynamics and video-text alignment. In particular, it uses a convolution projection and feature refinement module to better align video-text feature, the bi-directional cross-modal fusion network to capture query-aware clip feature, and the unidirectional joint-task feedback mechanism to strenghthen task correlation. Authors conduct lots of experiments on three benchmarks to examine the performance of the proposed method.

**Strengths:**

1) The idea is nice although combining several modules.
2) The experiments are convincing and the implementations are provided.
3) This paper is well structured and easy to read.

**Weaknesses:**

1) In Sec. 3.1, it simply adopts the concatenation to fuse the features extracted by SLOWFAST, CLIP, and BLIP. What about other fusion methods?
2) In Sec. 5, it claims that the model has fewer learnable parameters, and it is expected to compare the model size and the computational cost (GFLOPs).
3) In Fig. 1, the Class Prediction Head and the Localization Prediction Head are two different prediction heads, but their outputs are the same. Do the matrices $M$ have the same meaning?
4) In Sec. 3.3, it mentions that BI-CMF applies self-attention after cross-attention to extract the refined features, which is not shown in Figure 3. In addition, some ablations are required to show the influence of the self-attention layer.
5) In Table 1 and Table 2, the explanation should be given since in some cases the performance is worse than that using pre-training, e.g., 51.95 vs 51.56 in terms of R1@0.7 on QVHighlights，as well as VT, VU, DS methods on TVSum.

**Questions:**

See the weakness.

---

### Note · Authors · 2024-11-28

I have read and agree with the venue's withdrawal policy on behalf of myself and my co-authors.